# Association of Leukocyte, Erythrocyte, and Platelet Counts with Metabolic Syndrome and Its Components in Young Individuals without Overt Signs of Inflammation: A Cross-Sectional Study

**DOI:** 10.3390/children11010066

**Published:** 2024-01-04

**Authors:** Katarína Šebeková, Radana Gurecká, Ľudmila Podracká

**Affiliations:** 1Institute of Molecular BioMedicine, Faculty of Medicine, Comenius University, 81108 Bratislava, Slovakia; radana.gurecka@fmed.uniba.sk; 2Institute of Medical Physics, Biophysics, Informatics and Telemedicine, Faculty of Medicine, Comenius University, 81372 Bratislava, Slovakia; 3Department of Pediatrics, Faculty of Medicine, National Institute for Children Health, Comenius University, 83340 Bratislava, Slovakia; ludmila.podracka@fmed.uniba.sk

**Keywords:** metabolic syndrome, sex difference, blood count, adolescents, continuous metabolic syndrome score, decision-tree model

## Abstract

The presence of metabolic syndrome (MetS) increases the risk of developing type 2 diabetes, cardiovascular diseases, and mortality. MetS is associated with increased leukocyte or erythrocyte counts. In 16- to 20-year-old males (n = 1188) and females (n = 1231) without signs of overt inflammation, we studied whether the presence of MetS and its components results in elevated blood cell counts. The leukocyte, erythrocyte, and thrombocyte counts significantly but weakly correlated with the continuous MetS score, MetS components, uric acid, and C-reactive protein levels both in males (r = −0.09 to 0.2; *p* < 0.01) and females (r = −0.08 to 0.2; *p* < 0.05). Subjects with MetS had higher leukocyte (males: 6.2 ± 1.3 vs. 6.9 ± 1.2 × 10^9^/L; females 6.6 ± 1.5 vs. 7.5 ± 1.6 × 10^9^/L; *p* < 0.001), erythrocyte (males: 5.1 ± 0.3 vs. 5.3 ± 0.3 × 10^12^/L; females: 4.5 ± 0.3 vs. 4.8 ± 0.3 × 10^12^/L; *p* < 0.001), and platelet counts (males: 245 ± 48 vs. 261 ± 47 × 10^9^/L; females: 274 ± 56 vs. 288 ± 74 × 10^9^/L; *p* < 0.05) than those without MetS. With the exception of platelet counts in females, the blood counts increased with the number of manifested MetS components. Phenotypes with the highest average leukocyte, erythrocyte, or platelet counts differed between sexes, and their prevalence was low (males: 0.3% to 3.9%; females: 1.2% to 2.7%). Whether functional changes in blood elements accompany MetS and whether the increase in blood counts within the reference ranges represents a risk for future manifestation of cardiometabolic diseases remain unanswered.

## 1. Introduction

Metabolic syndrome (MetS) is defined as the presence of at least three out of five cardiometabolic risk factors, e.g., central obesity, impaired glucose metabolism, elevated blood pressure (BP), elevated triacylglycerols (TAG), and low high-density lipoprotein cholesterol (HDL-C) concentrations [1]. The presence of MetS increases the probability of developing serious health conditions, including type 2 diabetes, cardiovascular diseases, or some types of cancer, eventually leading to increased mortality [2,3]. Although a few decades ago, MetS was considered exclusive to adults, worldwide epidemics of obesity contribute to its rising prevalence in youth [4,5].

Apart from the components of MetS, certain conditions and cardiometabolic risk markers, such as low-grade inflammation, elevated uric acid levels, or changes in blood counts, typically exist alongside MetS. MetS represents a state of prolonged subclinical inflammation crucial to the progression of cardiovascular disease [6,7]. Children and adolescents with MetS display higher levels of C-reactive protein (CRP), more frequently present with CRP > 3.0 mg/L compared with their MetS-free peers, and CRP levels associate with the components of MetS [8,9,10,11,12]. Similarly, leukocyte counts are higher in juveniles with MetS than those without; leukocyte counts rise with the increasing number of MetS components and are directly associated with fasting plasma glucose (FPG), insulin resistance, TAG levels, BP, and measures of obesity [10,13,14,15]. It has been suggested that the neutrophils-to-lymphocytes ratio (NLR) be used as a marker of low-grade inflammation when leukocyte counts are within the reference range [16]. However, data on the association of NLR with MetS are inconsistent [17,18,19]. Simultaneous evaluation of erythrocyte and leukocyte counts in adolescents using Bayesian modeling indicated their correlation with waist-to-height ratio (WHtR), BP, FPG, and TAG levels, while only erythrocyte counts correlated significantly with HDL-C levels [20]. Erythrocyte counts are higher in children with prehypertension and hypertension than in their normotensive peers [21]. Erythropoiesis is a constant source of endogenous uric acid [22]. Thus, higher erythrocyte counts are associated with higher uricemia [23]. In adolescents, uric acid levels correlate with measures of obesity, BP, glucose homeostasis, lipid profile, inflammatory markers, or hematologic variables, and clustering of cardiometabolic risk factors with rising uricemia imposes increased cardiometabolic risk [23,24,25,26]. Limited information exists regarding the association between MetS and its individual components with platelet counts in young individuals, and data on the prevalence of MetS across the quartiles of platelet counts are contradictory [27,28].

To our knowledge, no study has simultaneously addressed the association of blood counts with cardiometabolic risk factors and markers. The objective of the present study is to investigate the association between leukocyte, erythrocyte, and platelet counts and NLR with MetS, its components, uric acid, and CRP levels and to identify phenotypes associated with the highest blood counts in young adults without overt signs of inflammation. Considering sexual dimorphism in blood counts and in the pathogenesis and prevalence of MetS components [29], a separate evaluation was conducted for males and females. We anticipated that the presence of MetS associates in both sexes with higher blood counts and similarly affects cardiometabolic risk factors and markers, while phenotypes presenting with the highest blood counts differ between males and females.

## 2. Materials and Methods

The cross-sectional survey “Respect for Health” was conducted in the school year 2011/2012 at the state-run secondary schools in the Bratislava Region of Slovakia. The goal was to gather information on the cardiometabolic status of youth to implement adequate preventive measures. Enrollment in the study was voluntary and contingent upon the written informed consent of participants of full age or the signed consent of parents or guardians, along with verbal assent from participants under 18 years old. Exclusion criteria encompassed any acute or chronic illnesses, pregnancy, or lactation in females. The study protocol received approval from the Ethics Committee of the Bratislava Self-Governing Region.

From the database of 11- to 23-year-old students, data from 16- to 20-year-olds were extracted for current analysis. Following the exclusion of individuals with incomplete data, those with CRP levels exceeding 10 mg/L, and leukocyte counts > 10^9^/L, the final analysis comprised 2419 individuals (1231 females; 50.9%).

### 2.1. Measurements

Blood pressure and anthropometric measurements (including body weight, height, and waist circumference) were conducted at schools by trained personnel following a standardized protocol, as detailed previously [30]. BP was assessed using a digital monitor (Omron M-6 Comfort, Kyoto, Japan) on the right arm while seated, with a 10-minute rest period. The recorded value was the mean of the last two out of three measurements.

Following an overnight fast, blood samples were collected from the antecubital vein between 7 and 9 a.m. at designated medical centers. Fasting plasma glucose (FPG), lipid profile, and uric acid levels were assessed in the certified central laboratory using standard laboratory methods, employing the Advia 2400 analyzer (Siemens, Munich, Germany). High-sensitivity CRP and fasting plasma insulin (FPI) were quantified via immunoassay with direct chemiluminescence testing methodology (Advia Centaur XP Immunoassay System, Siemens, Germany). The Sysmex XE-2100 analyzer (Sysmex Corporation, Kobe, Japan) was utilized for blood count determinations.

### 2.2. Calculations and Classifications

The waist-to-height ratio (WHtR) and NLR were calculated. Insulin sensitivity was estimated using the quantitative insulin sensitivity check index (QUICKI) [31], the atherogenic index of plasma (AIP) was calculated as log((TAG)/HDL-C) [32], and the continuous metabolic syndrome score (cMSS)—an estimate of cardiometabolic risk—was derived by incorporating the components of MetS as follows: WHtR/0.5 + fasting glycemia/5.6 + SBP/130 + triacylglycerols/1.7 − HDL-C/1.02 (males) or 1.28 (females) [33]. WHtR, systolic BP (SBP), FPG, HDL-C, and TAG were considered MetS components, and uric acid and CRP were classified as cardiometabolic risk markers. To define adverse levels of FPG, SBP, HDL-C, and TAG, cut-off values of the International Diabetic Federation were used [1]. Central obesity was defined as WHtR ≥ 0.5 [34], insulin resistance as FPI > 20 µIU/mL [35], hyperuricemia (according to the age- and sex-specific reference ranges of the laboratory) as ≥340 µmol/L in females (F) and ≥420 µmol/L in males (M). CRP > 3 mg/L was considered an indicator of increased cardiovascular risk [36], and AIP ≥ 0.11 was an indicator of increased atherogenic risk [32]. MetS was defined as the presence of at least three conditions, e.g., SBP ≥ 130 mmHg or diastolic BP ≥ 85 mmHg; TAG ≥ 1.7 mmol/L; FPG ≥ 5.6 mmol/L; WHtR ≥ 0.5; HDL-C: <1.03 mmol/L (M) and <1.29 mmol/L (F).

### 2.3. Statistical Analyses

Comparison of continuous variables between males and females was conducted using the two-sided Student’s *t*-test for independent samples, with Levene’s test employed to assess the homogeneity of variances. The importance of the difference was assessed by calculation of effect size. Cohen’s *d* < 0.2 was considered unimportant; values for small, medium, and large effect sizes were set at 0.2, 0.5, and 0.8, respectively. Calculation of Pearson’s correlation coefficients was performed. To assess the influence of sex, the presence or absence of the MetS, and their interaction with dependent variables, a two-factor analysis of variance (ANOVA) was employed. Trends of continuous variables (MetS components and cardiometabolic risk markers) across the subgroups according to the presented number of MetS components were evaluated using the one-way ANOVA.

To predict outcomes and illustrate the interaction patterns among independent variables affecting blood counts, a decision-tree model utilizing chi-squared automatic interaction detection (CHAID) was employed. The model incorporated independent dichotomized variables (e.g., WHtR, SBP, FPG, FPI, TAG, HDL-C, CRP, and uric acid; categorized above/below the cut-off values), along with age as an influence variable. Normally distributed data are presented as mean ± standard deviation (SD). Skewed data were log-transformed before the analyses and are reported as back-transformed geometric mean (−1 SD; 1 SD). The statistical analyses were performed using SPSS software (v. 16 for Windows, SPSS, Chicago, IL, USA). A *p*-value < 0.05 was considered statistically significant.

## 3. Results

### 3.1. General Characteristics of the Subjects

Cohort characteristics are given in Table 1. Considering the effect size, males differed from females in all variables except age, FPI concentration, QUICKI, TAG levels, and lymphocyte counts.

Among males, 6.3% presented with MetS, while the prevalence in females reached 1.3% (*p* < 0.001). Compared with males, females showed a lower prevalence of central obesity (12.9% vs. 9.0%, *p* = 0.003), elevated SBP (M: 25.8%; F: 1.4%), FPG (M: 6.6%; F: 2.0%), AIP (M: 8.2%; F: 3.7%), and uric acid levels (M: 13.0%; F: 5.4%), while that of low HDL-C (M: 13.8%, F: 22.5%) and elevated CRP concentration (M: 6.1%; F: 10.2%) was higher in females (*p* < 0.001, all). No significant sex difference was revealed in the prevalence of elevated TAG (M: 5.1%, F: 5.0%; *p* = 1.000) or FPI levels (M: 4.5%, F: 3.3%; *p* = 0.171).

### 3.2. Association between Components of MetS, Continuous MetS Score, and Hematologic Variables

Pearson correlation coefficients were computed to investigate the relationship between hematologic variables and individual components of MetS or markers of cardiometabolic risk (Table 2). The number of leukocytes, neutrophils, erythrocytes, and platelets significantly correlated with most cardiometabolic risk factors and markers in both sexes. However, all significant Pearson correlation coefficients were low, reaching a maximum of 0.224 in males and 0.197 in females. The leukocyte and erythrocyte counts (M: r = 0.154; F: r = 0.157), as well as the leukocyte and platelet counts (M: r = 0.264; F: r = 0.311) correlated significantly (*p* < 0.001, all) in both sexes. No significant correlation was revealed between the erythrocyte and platelet counts (M: r = −0.004, *p* = 0.887; F: r = 0.056, *p* = 0.050).

### 3.3. Cardiometabolic Variables in Subjects with and without MetS

To compare the health impacts of MetS in males and females, we ran a two-way ANOVA (Table 3). The presence of MetS adversely affected all variables but lymphocyte counts and NLR. Sex had no significant impact on WHtR, QUICKI, AIP, TAG, CRP levels, and lymphocyte counts. Two-way ANOVA did not indicate sex–MetS interaction in any case.

Next, we explored the trends of hematologic variables and markers of cardiometabolic risk across subgroups based on the number of MetS components. Among males, 53.9% did not show any component of MetS, 26.1% displayed one, 13.9% presented with two components, and 6.1% suffered from MetS (Table 4). All variables showed significant trends across the subgroups. Individuals suffering from MetS displayed the least favorable values in all metabolic indicators. F values for trends in blood cell counts and NLR were lower than those for cardiometabolic risk markers (CRP, uricemia).

Among females, 65.1% did not present any MetS component, 27.3% displayed one, 6.3% showed two components, and 1.3% suffered from MetS (Table 5). Unlike in males, the trend in NLR and platelet counts did not reach significance in females. F values for CRP were higher than those for white blood cell counts, while those for erythrocyte counts and uricemia were almost identical.

### 3.4. Decision-Tree Model

The decision-tree model was utilized to forecast the impact of MetS components, cardiometabolic risk markers, and their interaction with blood cell counts. In the primary split, CRP was chosen as the predictor variable for leukocyte counts in males (Table 6) and females (Table 7). In males with CRP levels ≤ 3 mg/L, WHtR, and subsequently TAG levels in lean subjects and FPI levels in centrally obese ones significantly determined leukocyte counts. A rare phenotype (node 8) with CRP ≤ 3 mg/L (mean: 1.4 (0.7; 2.7 mg/L)), presenting with central obesity (WHtR: 0.56 ± 0.04) and insulin resistance (FPI: 39.7 (29.2; 54.0 µIU/mL)) displayed the highest mean leukocyte counts. The second-highest counts were revealed in males whose CRP values were >3 mg/L. Among females, in those with elevated CRP, HDL-C appeared as a classifier in the second node, while WHtR predicted leukocyte counts in those with low CRP levels and TAG category in the subsequent node in lean females. Females concurrently displaying CRP values > 3 mg/L (mean: 4.2 (3.1; 5.6 mg/L)) and low HDL-C (1.07 ± 0.14 mmol/L) had the highest leukocyte counts, followed by their peers presenting elevated CRP (4.8 (3.3; 6.8 mg/L)) and HDL-C above 1.29 mmol/L (1.67 ± 0.27 mmol/L). These phenotypes were presented by 2.7% and 7.5% of females, respectively.

Centrally obese insulin-resistant males with CRP levels below 3 mg/L displayed the highest mean neutrophil counts (4.3 ± 1.2 × 10^9^/L). The prevalence of this phenotype reached 1.9%. Among females, 0.2% characterized as lean with elevated FPG and CRP levels showed the highest mean neutrophil count (6.0 ± 1.9 × 10^9^/L). In both sexes, subjects with elevated TAG levels displayed the highest lymphocyte counts (males: 2.1 ± 0.5 × 10^9^/L, prevalence: 5.1%; females: 2.2 ± 0.5 × 10^9^/L, prevalence: 5.0%). Out of all males, 6.1% with CRP levels above 3 mg/L displayed the highest mean NLR (2.5 ± 1.1). Among females, 0.2% displaying normotriacylglycerolemia, elevated CRP, and elevated FPG levels showed the highest mean NLR (3.6 ± 2.2).

WHtR appeared at the decision tree’s top-level node, classifying erythrocyte counts in both sexes (M: Table 6; F: Table 7). A rare phenotype (prevalence: 0.3%), characterized by central obesity (WHtR: 0.53 ± 0.03), insulin sensitivity (FPI: 14.9 (11.9; 18.8 µIU/mL)), and elevated FPG (5.9 ± 0.3 mmol/L) displayed the highest mean erythrocyte counts, while centrally obese (WHtR: 0.56 ± 0.04) insulin-resistant (FPI: 40.5 (28.9; 56.6 µIU/mL)) males presented with the second-highest mean counts. The latter phenotype was presented by 2.6% of males. In males without central obesity, TAG levels predicted erythrocyte counts in the second split, and in those not displaying elevated TAG levels, SBP was a predictor. Centrally obese females (WHtR: 0.55 ± 0.05) with elevated uric acid levels (363 ± 20 mmol/L) and lean females (WHtR: 0.42 ± 0.04) displaying elevated FPG (5.8 ± 0.2 mmol/L) represented phenotypes with the highest erythrocyte counts. Their prevalence reached 1.3% and 1.7%, respectively. In lean normoglycemic females, HDL-C levels predicted erythrocyte counts.

Centrally obese males (WHtR: 0.55 ± 0.04) with hyperuricemia (uric acid: 461 ± 34 mmol/L) showed the highest platelet counts (Table 6). Those with central obesity (WHtR: 0.54 ± 0.04), normouricemia (uric acid: 359 ± 40 mmol/L), and normoglycemia (4.9 ± 0.3 mmol/L) displayed the second-highest counts. In total, 3.9% and 8.2% of males presented with these phenotypes. In lean males, CRP levels, and subsequently in those with CRP ≤ 3 mg/L, HDL-C levels predicted thrombocyte counts. Normotriacylglycerolemic females (0.71 ± 0.30 mmol/L) with elevated SBP (133 ± 3 mm Hg) showed the highest platelet counts (Table 7), while those with elevated TAG levels (2.09 ± 0.44 mmol/L) displayed the second-highest values. The prevalence of the mentioned phenotypes was 1.2% and 5.0%, respectively. The decision tree indicated that CRP significantly determines platelet counts in normotriacylglycerolemic females without elevated SBP.

## 4. Discussion

This study provides evidence that young males and females with MetS who do not display overt signs of inflammation present with higher leukocyte, neutrophil, erythrocyte, and platelet counts than their MetS-free peers. Leukocyte, neutrophil, and erythrocyte counts show an increasing trend with the rising number of MetS components in both sexes, while the NLR and platelet counts increase across the categories only in males. Both sexes also display worsening trends in components of MetS and cardiometabolic risk markers with an increasing number of MetS components. Observed changes in blood counts seem similar to those for standard cardiometabolic risk markers—uricemia or CRP levels—in females and are perhaps slightly less expressed in males. Generally, phenotypes with a higher risk show higher mean blood counts. Phenotypes displaying the highest mean leukocytes, neutrophils, NLR, erythrocytes, or platelet counts differ in males and females, and their prevalence is low. Differences in blood counts were manifested within their reference ranges. Unless it is proven that the rising number of blood elements within the reference range represents an increased risk for future manifestation of cardiovascular diseases, our finding is of limited implication for clinical practice.

The prevalence of MetS in our cohort reached 3.7%. This is similar to that reported for Chinese adolescents (3%) [37] but lower than indicated in a general population of Korean (6.7%) [13], white American (15.2%) [38], or severely obese Italian adolescents (26.0%) [17]. Only a few studies on young subjects reported sex-specific prevalence. In most studies, the prevalence was higher in males than females [13,33,36], albeit an opposite ratio has also been reported [28]. According to Ostrihonova et al. [39], the prevalence of MetS in 18- to 25-year-old Slovaks reached 7% in males and 4% in females. Our males displayed a similar prevalence, while MetS was less frequent in our females. In line with their data [39], elevated blood pressure emerged as the most common component of MetS in our males, while low HDL-C was predominant in females. The comparison of the prevalence reported by different groups is to be interpreted with caution, as there is neither a consensus on cut-offs for MetS components nor on rules of MetS classification (e.g., any three components, central obesity + any other two components, insulin resistance + any other two components) [40,41,42,43]. Thus, the reported prevalence of MetS depends on the guidelines employed.

Leukocyte and platelet counts are higher, while erythrocyte counts are lower in females than in males [44,45,46]. Differences in leukocyte counts are attributed to differences in sex hormones, which influence immune function at different levels [44]. This results in greater immune responsiveness in females than males, mirrored, among others, by higher CRP levels in females [47]. Sexual dimorphism in erythrocyte counts has been attributed to periodic menstrual blood losses in females of reproductive age and the fact that higher testosterone levels in males facilitate erythropoiesis [48]. Since these mechanisms are not mirrored by different physiological concentrations of erythropoietin in males and females, Murphy et al. [49] proposed that the lower requirement for erythrocytes in females reflects their more efficient delivery to the capillary circulation. Despite the sex differences in leukocyte and erythrocyte counts, our subjects with MetS displayed higher counts than their MetS-free peers, regardless of sex. Devesa et al. [50] observed bone marrow activation in apparently healthy middle-aged adults presenting with MetS, even in the absence of systemic inflammation. This activation was reflected in elevated circulating leukocyte and erythrocyte counts and showed an association with the components of MetS and atherosclerosis.

Among several studies focusing on the association between leukocyte counts and MetS in adolescents, only two compared the counts between MetS-free and MetS-positive subjects. Neither reported significant between-group differences, albeit in both studies, MetS-free groups presented with more favorable measures of all MetS components [17,51]. As the Italian study dealt with a clinical sample of obese adolescents [17], it might be speculated that obesity per se triggers a rise in leukocyte counts within a normal range to such an extent that additional MetS components have only a negligible effect. On the other hand, Hsieh et al. [51] studied a general group of Taiwanese adolescents with leukocyte counts within a reference range. Whether genetic, lifestyle, or other factors underlie the difference remains unclear. However, our data are in line with other studies on adolescents, indicating that leukocyte counts significantly correlate with cardiometabolic risk factors and markers, such as measures of obesity, BP, FPG, HDL-C, TAG, insulin, uric acid, or CRP levels [15,23,38,52], and show increasing trends as the number of MetS components increases [13,38]. Importantly, we also show that blood cell counts correlate not only with single cardiometabolic risk factors and markers but also with MetS severity (evaluated as cMSS).

MetS is a state of chronic low-grade inflammation mirrored, among others, by the rise of leukocyte counts and CRP levels within their respective reference range. CRP is an acute phase reactant with a short half-life, thus representing a more sensitive indicator of low-grade inflammation than the leukocyte count. In adults, CRP levels correlate with all components of MetS and uric acid levels [53], and high CRP levels within the reference range increase the risk of cardiovascular events beyond that imparted by the other metabolic risk factors [54]. The exact link between MetS and CRP is not fully understood. CRP levels are increased particularly in obese insulin-resistant subjects. In obesity, visceral fat tissue releases proinflammatory cytokines into the portal vein, directly triggering CRP production and boosting an oxidative stress response in the liver. These, in turn, contribute to the development of insulin resistance and atherogenic dyslipidemia [55]. On the other hand, the relation between CRP concentrations and insulin resistance or reactive hyperinsulinemia is independent of obesity [56]. The modulatory effect of low-grade inflammation on leukocyte counts in apparently healthy young adults is reflected by the fact that the decision-tree model selected CRP in the first split in both sexes. Low HDL-C levels in females with CRP > 3 mg/L were associated with the highest leukocyte counts among the end-node phenotypes. In adolescents and adults, HDL-C levels show an inverse relationship with leukocyte counts [38,57]. The impact of low HDL-C is not surprising, as HDL has anti-inflammatory and antiatherogenic effects [58,59]. As formerly shown, total peripheral leukocyte counts are directly associated with TAG levels in adolescents and adults [15,20,38,57]. In our study, TAG levels modulated leukocyte counts in lean adolescents with CRP levels < 3 mg/L. In males, elevated CRP levels impacted leukocyte counts similarly to central obesity combined with insulin resistance in subjects with low CRP levels. In our study, CRP levels and leukocyte (neutrophil) counts were better indicators of MetS-associated low-grade inflammation than NLR. Similar to the Chinese study on apparently healthy adults with CRP < 10 mg/L [60], we did not reveal a significant difference in NLR between subjects with and without MetS. The trend of NLR across the rising number of MetS components was significant (but mild and nonlinear) only in our males. The Italian study reported that obese adults with MetS, but not obese adolescents, display higher NLR than their MetS-free peers and that NLR rises with the increasing number of MetS components [17]. In another Chinese study [61], adults with MetS had higher NLR than those without, but NLR did not increase significantly with the rising number of MetS components. Taken together, available data do not support the assumption that in MetS NLR is a suitable indicator of chronic low-grade inflammation if the leukocyte count is within the reference range.

In adults with MetS, higher erythrocyte counts and their significant associations with MetS components [23,57] or MetS-associated anemia resulting from low-grade inflammation-induced iron deficiency [61,62] have been described. Data on young subjects are scarce. Consistent with the findings of Mansourian et al. [20], we revealed higher erythrocyte counts in individuals with MetS than in those without, and these counts increased with the worsening of MetS components. Interestingly, the decision-tree model identified uric acid as the predictor variable for the erythrocyte count only in females. During erythropoiesis, nuclei are extruded from erythroblasts; thus, nuclear disintegration represents a constant source of endogenous uric acid. We anticipated that hyperuricemia would serve as a predictive factor for erythrocyte counts, particularly in males, given their higher erythrocyte counts and uric acid levels compared to females. Furthermore, the prevalence of hyperuricemia in males in our study was more than twice that in females. In contrast to the reports in adults on MetS-associated anemia [61,62], we revealed the lowest mean erythrocyte counts in males presenting with the most favorable phenotype (Table 6). In the last node, females with low HDL-C had significantly lower mean erythrocyte counts than those with levels > 1.29 mmol/L (Table 7). Still, the difference is of statistical rather than clinical importance.

The relationship between platelet counts and MetS and its components in adolescents is seldom explored. Increased platelet counts have been linked to a higher prevalence and risk of MetS [28], and platelet counts rose with increasing BMI [52,63] and insulin resistance [14]. Our finding that platelet counts show the highest correlation with those of leukocytes and CRP levels aligns with their role in inflammation and atherosclerosis [64]. Albeit individuals with MetS in our study exhibited higher platelet counts than those without MetS, platelet counts increased with a rising number of MetS components only in males. The data for adults also did not prove a convincing relationship between platelet count and MetS. Fang et al. [65] concluded that platelet counts are significantly higher in MetS-positive vs. MetS-free Taiwanese males and females. However, they compared thousands of probands, and the between-group means differed by 2 × 10^9^/L and 3 × 10^9^/L, respectively, yielding an effect size (Cohen’s *d*) of 0.05 and 0.03. In another large study on Chinese by Li et al. [66], no significant differences between the groups with and without MetS were revealed. While the overall trend in platelet counts across groups with an increasing number of MetS components was significant, the location of differences was random.

As observed increases in leukocyte, erythrocyte, and platelet counts occur within their reference ranges, clinical implications of our finding of higher counts of blood elements in the presence of MetS and rising trends with an increasing number of MetS components may appear negligible at first glance. However, the question arises whether an increase in leukocyte, erythrocyte, or platelet counts within their reference range might be of prognostic value. Research conducted on nondiabetic Korean juveniles with leukocyte counts within the reference range indicates that when comparing the fourth to the first quartile of leukocyte counts, the odds for insulin resistance increase threefold [14], and those for MetS are fivefold higher in males and sevenfold higher in females [13]. Males in the fourth vs. the first quartile of platelet counts displayed 5-fold higher and females had 4-fold higher odds for MetS [28]. A longitudinal study involving initially healthy Chinese children aged 6 to 8 years revealed that at a follow-up conducted 5 years later, the incidence of prehypertension and hypertension was 3.3-fold higher in those with erythrocyte counts within the fourth quartile vs. the first quartile at baseline [21]. Longitudinal studies in adults document that a rise in leukocyte, erythrocyte, or platelet counts within their normal range is associated with an increased incidence of type 2 diabetes, cardiovascular diseases, and mortality later (reviewed in [16]). Moreover, functional changes of blood elements in MetS have been described [55,67,68,69].

The strength of our study lies in the inclusion of a reasonably large cohort of young participants without overt signs of inflammation and simultaneous assessment of leukocyte, erythrocyte, and platelet counts in conjunction with cardiometabolic risk factors and markers. This information is essential for understanding the early phases of the association of MetS (a cardiometabolic risk factor), blood counts, and markers of low-grade inflammation. Another strength is the endeavor to characterize phenotypes associated with high counts of blood elements. The main limitation of our study lies in its cross-sectional design, preventing the assessment of causality; instead, we can only draw associations. We do not have data on genetics, family history of chronic degenerative diseases, or lifestyle factors, such as dietary regimen or physical activity, all of which can influence the manifestation of MetS components and blood counts. Our data are based on single measurements and blood samplings from each subject. Our examination was limited to students enrolled in state secondary schools within the Bratislava Region, who voluntarily took part in the survey. Thus, our results do not apply to other populations. Participants were young, and the prevalence of MetS and its components was relatively low. On the other hand, data were not affected by comorbidities occurring at higher age. Despite these limitations, our findings offer insight into the hematologic changes in young individuals with pre-MetS and MetS.

## 5. Conclusions

Longitudinal studies are essential to determine whether an increase in blood counts within the reference range during adolescence may serve as an early indicator of future risk of developing cardiovascular diseases as similarly documented for an increase in CRP levels. The attempt to objectivize this challenge is attractive, as a complete blood count is a routine and cheap analysis.

## Figures and Tables

**Table 1 children-11-00066-t001:** Cohort characteristics.

	Males (n = 1188; 49.1%)	Females (n = 1231; 50.9%)	*p*	Cohen’s *d*
Age, years	17.4 ± 1.2	17.5 ± 1.1	0.060	--
WHtR	0.44 ± 0.05	0.43 ± 0.05	**<** **0.001**	**0.2**
SBP, mm Hg	123 ± 12	107 ± 9	**<** **0.001**	**1.5**
FPG, mmol/L	4.9 ± 0.4	4.7 ± 0.4	**<** **0.001**	**0.5**
FPI, µIU/mL	9.4 (5.6; 15.8)	9.9 (6.2; 15.8)	**0.025**	0.1
QUICKI	0.344 ± 0.028	0.344 ± 0.025	0.956	--
HDL-C, mmol/L	1.25 ± 0.23	1.52 ± 0.30	**<** **0.001**	**1.0**
TAG, mmol/L	0.79 (0.51; 1.21)	0.81 (0.53; 1.22)	0.220	--
AIP	−0.19 ± 0.22	−0.27 ± 0.20	**<** **0.001**	**0.4**
cMSS	2.01 ± 0.47	1.86 ± 0.39	**<** **0.001**	**0.3**
Uric acid, mmol/L	354 ± 59	257 ± 51	**<** **0.001**	**1.8**
CRP, mg/L	0.4 (0.1; 1.4)	0.5 (0.2; 1.8)	**0.001**	**0.2**
Leukocytes, ×10^9^/L	6.3 ± 1.3	6.6 ± 1.5	**<** **0.001**	**0.2**
Neutrophils, ×10^9^/L	3.7 ± 1.0	4.2 ± 1.2	**<** **0.001**	**0.4**
Lymphocytes, ×10^9^/L	1.92 ± 0.49	1.87 ± 0.48	**0** **.015**	0.1
NLR	2.1 ± 0.9	2.4 ± 0.9	**<** **0.001**	**0.3**
Erythrocytes, ×10^12^/L	5.1 ± 0.3	4.6 ± 0.3	**<** **0.001**	**1.7**
Platelets, ×10^9^/L	246 ± 48	274 ± 56	**<** **0.001**	**0.5**

WHtR, waist-to-height ratio; SBP, systolic blood pressure; FPG, fasting plasma glucose; FPI, fasting plasma insulin; QUICKI, quantitative insulin sensitivity check index; HDL-C, high-density lipoprotein cholesterol; TAG, triacylglycerols; AIP, atherogenic index of plasma; cMSS5, continuous metabolic syndrome score; CRP, C-reactive protein; NLR, neutrophils-to-lymphocytes ratio; data were compared using the two-sided Student’s *t*-test for independent groups; normally distributed data are given as mean ± standard deviation (SD); data not fitting to normal distribution were log-transformed and are given as back-transformed geometric mean (−1 SD, +1 SD); *p* < 0.05 was considered significant and is given in bold; --, not calculated; effect size (Cohen’s *d*) < 0.2 was considered unimportant.

**Table 2 children-11-00066-t002:** Pearson correlations between hematologic variables and metabolic syndrome components or cardiometabolic risk markers in males and females.

	**Males**					
	**Leukocytes**	**Neutrophils**	**Lymphocytes**	**NLR**	**Erythrocytes**	**Platelets**
WHtR	**0.150 *****	**0.150 *****	**0.074 ****	0.039	**0.141 *****	**0.131 *****
SBP	0.051	**0.076 ****	0.012	**0.057 ***	**0.146 *****	−0.004
FPG	**0.067 ***	**0.060 ***	−0.027	0.017	**0.061 ***	**0.076 ****
LnFPI	**0.112 *****	**0.111 *****	**0.076 ****	**0.058 ***	**0.173 *****	**0.128 *****
QUICKI	**−0.106 ***	**−0.104 *****	**−0.060 ***	−0.049	**−0.175 *****	**−0.128 *****
HDL-C	**−0.090 ****	**−0.082 ****	−0.013	−0.037	**−0.090 ****	0.015
LnTAG	**0.215 *****	**0.161 *****	**0.193 *****	0.001	**0.224 *****	**0.086 ****
AIP	**0.210 *****	**0.162 *****	**0.178 *****	0.013	**0.217 *****	**0.069 ***
cMSS	**0.205 *****	**0.173 *****	**0.142 *****	0.032	**0.222 *****	**0.086 ****
Uric acid	**0.073 ***	0.044	0.050	−0.018	**0.094 ****	0.045
LnCRP	**0.220 *****	**0.221 *****	0.054	**0.128 *****	0.032	**0.153 *****
	**Females**					
	**Leukocytes**	**Neutrophils**	**Lymphocytes**	**NLR**	**Erythrocytes**	**Platelets**
WHtR	**0.169 *****	**0.165*****	**0.074 ****	**0.075 ****	**0.082 ****	**0.085 ****
SBP	0.010	0.008	0.012	−0.017	**0.145 *****	0.028
FPG	0.009	0.029	−0.027	0.043	**0.065 ***	0.053
LnFPI	**0.185 *****	**0.183 *****	**0.076 ****	**0.098 ****	**0.113 *****	**0.090 ****
QUICKI	**−0.174 *****	**−0.177 *****	**−0.060 ***	**−0.103 *****	**−0.121 *****	**−0.087 ****
HDL-C	**−0.084 ****	**−0.081 ****	−0.013	**−0.060 ***	−0.015	0.026
LnTAG	**0.174 *****	**0.122 *****	**0.193 ****	−0.017	**0.092 ****	**0.125 ****
AIP	**0.197 *****	**0.150 *****	**0.178 *****	0.016	**0.091 ****	**0.100 *****
cMSS	**0.193 *****	**0.161 *****	**0.142 *****	0.039	**0.112 *****	**0.104 *****
Uric acid	**0.128 *****	**0.121 *****	0.050	**0.062 ***	**0.124 *****	0.038
LnCRP	**0.197 *****	**0.196 *****	0.054	**0.128 *****	0.013	**0.135 *****

NLR, neutrophils-to-lymphocytes ratio; WHtR, waist-to-height ratio; SBP, systolic blood pressure; FPG, fasting plasma glucose; Ln, logarithm; FPI, fasting plasma insulin; QUICKI, quantitative insulin sensitivity check index; HDL-C, high-density lipoprotein cholesterol; TAG, triacylglycerols; AIP, atherogenic index of plasma; cMSS, continuous metabolic syndrome score; CRP, C-reactive protein; *: *p* < 0.05; **: *p* < 0.01; ***: *p* < 0.001. Significant Pearson correlations are given in bold.

**Table 3 children-11-00066-t003:** Characteristics of the study population stratified by sex and presence or absence of MetS.

	Males		Females		*p*		
	Non-MetS (n = 1115)	MetS (n = 73)	Non-MetS (n = 1215)	MetS (n = 16)	Sex	MetS	S*MetS
Age, years	17.4 ± 1.1	17.8 ± 1.2	17.5 ± 1.1	18.1 ± 1.2	0.292	**0.001**	0.707
WHtR	0.44 ± 0.04	0.52 ± 0.06	0.43 ± 0.04	0.53 ± 0.08	0.864	**<** **0.001**	0.336
SBP, mm Hg	121 ± 11	138 ± 12	107 ± 9	121 ± 8	**<** **0.001**	**<** **0.001**	0.301
FPG, mmol/L	4.9 ± 0.4	5.2 ± 0.5	4.7 ± 0.4	5.0 ± 0.6	**<** **0.001**	**<** **0.001**	0.557
FPI, µIU/mL	9.1 (5.6; 14.8)	16.7 (8.98 31.9)	9.8 (6.2; 15.5)	21.1 (12.4; 35.9)	**0.021**	**<** **0.001**	0.221
QUICKI	0.346 ± 0.026	0.316 ± 0.030	0.344 ± 0.025	0.307 ± 0.023	0.133	**<** **0.001**	0.266
HDL-C, mmol/L	1.26 ± 0.22	1.06 ± 0.22	1.53 ± 0.30	1.248 ± 0.23	**<** **0.001**	**<** **0.001**	0.268
TAG, mmol/L	0.77 (0.52; 1.14)	1.25 (0.68; 2.30)	0.80 (0.53; 1.20)	1.36 (0.91; 2.03)	0.260	**<** **0.001**	0.721
AIP	−0.21 ± 0.20	0.08 ± 0.32	−0.27 ± 0.20	0.05 ± 0.19	0.091	**<** **0.001**	0.672
cMSS	1.95 ± 0.38	2.89 ± 0.74	1.85 ± 0.37	2.77 ± 0.39	**0.046**	**<** **0.001**	0.834
Uric acid, mmol/L	351 ± 57	402 ± 68	257 ± 50	302 ± 47	**<** **0.001**	**<** **0.001**	0.681
CRP, mg/L	0.4 (0.1; 1.3)	0.9 (0.3; 2.9)	0.5 (0.2; 1.8)	1.0 (0.3; 3.2)	0.300	**<** **0.001**	0.866
Leukocytes, 10^9^/L	6.2 ± 1.3	6.9 ± 1.2	6.6 ± 1.5	7.5 ± 1.6	**0.005**	**<** **0.001**	0.476
Neutrophils, 10^9^/L	3.7 ± 1.0	4.1 ± 1.0	4.1 ± 1.2	5.0 ± 1.3	**<** **0.001**	**<** **0.001**	0.150
Ly, 10^9^/L	2.0 ± 0.5	2.1 ± 0.9	1.9 ± 0.5	1.9 ± 0.4	0.093	0.068	0.255
NLR	2.1 ± 0.9	2.1 ± 0.9	2.3 ± 0.9	2.7 ± 0.8	**<** **0.001**	0.078	0.221
Ery, 10^12^/L	5.1 ± 0.3	5.3 ± 0.3	4.5 ± 0.3	4.8 ± 0.3	**<** **0.001**	**<** **0.001**	0.064
Platelets, 10^9^/L	245 ± 48	261 ± 47	274 ± 56	288 ± 74	**<** **0.001**	**0.039**	0.961

MetS, metabolic syndrome; S*MetS, sex–metabolic syndrome interaction; WHtR, waist-to-height ratio; SBP, systolic blood pressure; FPG, fasting plasma glucose; FPI, fasting plasma insulin; QUICKI, quantitative insulin sensitivity check index; HDL-C, high-density lipoprotein cholesterol; TAG, triacylglycerols; AIP, atherogenic index of plasma; cMSS, continuous metabolic syndrome score; CRP, C-reactive protein; Ly, lymphocytes; NLR, neutrophils-to-lymphocytes ratio; Ery, erythrocytes. Data were compared using the two-way analysis of variance. Normally distributed data are given as mean ± standard deviation (SD). Data not fitting to normal distribution were log-transformed and are given as back-transformed geometric mean (−1 SD, +1 SD). *p* ≤ 0.05 was considered statistically significant and is shown in bold.

**Table 4 children-11-00066-t004:** The trends of hematologic variables and cardiometabolic risk factors and markers across subgroups based on the number of metabolic syndrome components presented in males.

MetS comp. No.	Leu, ×10^9^/L	Neu, ×10^9^/L	Ly, ×10^9^/L	NLR	Ery, ×10^12^/L	Plt, ×10^9^/L
0 (n = 640)	6.2 ± 1.2	3.6 ± 1.0	1.9 ± 0.5	2.0 ± 0.9	5.1 ± 0.3	246 ± 48
1 (n = 310)	6.2 ± 1.3	3.7 ± 1.0	1.9 ± 0.5	2.0 ± 0.7	5.2 ± 0.3	243 ± 48
2 (n = 165)	6.6 ± 1.4	4.0 ± 1.1	1.9 ± 0.5	2.3 ± 0.9	5.2 ± 0.3	248 ± 47
≥3 (n = 73)	6.9 ± 1.2	4.0 ± 1.0	2.1 ± 0.6	2.1 ± 0.9	5.3 ± 0.3	261 ± 47
F	10.2	11.8	3.6	3.0	10.8	2.7
*p*	**<0.001**	**<0.001**	**0.013**	**0.029**	**<0.001**	**0.045**
MetS comp. No.	WHtR	SBP, mm Hg	FPG, mmol/L	FPI, µIU/mL	QUICKI	HDL-C, mmol/L
0 (n = 640)	0.43 ± 0.03	117 ± 8	4.9 ± 0.3	8.2 (5.3; 12.7)	0.352 ± 0.024	1.31 ± 0.20
1 (n = 310)	0.45 ± 0.05	124 ± 10	5.0 ± 0.4	9.8 (5.9; 16.0)	0.342 ± 0.027	1.20 ± 0.23
2 (n = 165)	0.47 ± 0.06	134 ± 13	5.0 ± 0.4	12.0 (6.9; 20.6)	0.331 ± 0.026	1.20 ± 0.24
≥3 (n = 73)	0.52 ± 0.06	138 ± 12	5.2 ± 0.5	16.7 (9.9; 28.3)	0.316 ± 0.030	1.06 ± 0.22
F	158	158	20.6	66.1	61.5	43.9
*p*	**<0.001**	**<0.001**	**<0.001**	**<0.001**	**<0.001**	**<0.001**
MetS comp. No.	TAG, mmol/L	AIP	UA, mmol/L	cMSS	CRP, mg/L	
0 (n = 640)	0.76 ± 0.26	−0.26 ± 0.17	345 ± 56	1.79 ± 0.30	0.4 (0.1; 1.1)	
1 (n = 310)	0.87 ± 0.37	−0.17 ± 0.21	356 ± 60	2.08 ± 0.33	0.5 (0.2; 1.6)	
2 (n = 165)	1.05 ± 0.52	−0.10 ± 0.24	362 ± 54	2.32 ± 0.42	0.6 (0.2; 1.6)	
≥3 (n = 73)	1.51 ± 1.05	0.08 ± 0.32	402 ± 68	2.89 ± 0.74	0.9 (0.3; 2.9)	
F	81.6	79.5	23.6	263	15.8	
*p*	**<0.001**	**<0.001**	**<0.001**	**<0.001**	**<0.001**	

MetS, metabolic syndrome; No., number; Leu, leukocytes; Neu, neutrophils; Ly, lymphocytes; NLR, neutrophils-to-lymphocytes ratio; Ery, erythrocytes, Plt, platelets; WHtR, waist-to-height ratio; SBP, systolic blood pressure; FPG, fasting plasma glucose; PI, fasting plasma insulin; QUICKI, quantitative insulin sensitivity check index; HDL-C, high-density lipoprotein cholesterol; TAG, triacylglycerols; AIP, atherogenic index of plasma; UA, uric acid; cMSS, continuous metabolic syndrome score; CRP, C-reactive protein; F, the ratio of the between-group and within-group variation (if the null hypothesis is true, you expect F value close to 1). Data were compared using the analysis of variance. Normally distributed data are given as mean ± standard deviation (SD). Data not fitting to normal distribution were log-transformed and are presented as back-transformed geometric mean (−1 SD, +1 SD). *p* ≤ 0.05 was considered statistically significant and is shown in bold.

**Table 5 children-11-00066-t005:** The trends of hematologic variables and cardiometabolic risk factors and markers across subgroups according to the number of presented components of metabolic syndrome in females.

MetS comp. No.	Leu, ×10^9^/L	Neu, ×10^9^/L	Ly, ×10^9^/L	NLR	Ery, ×10^12^/L	Plt, ×10^9^/L
0 (n = 801)	6.5 ± 1.4	4.1 ± 1.2	1.8 ± 0.5	2.3 ± 0.9	4.5 ± 0.3	272 ± 53
1 (n = 336)	6.8 ± 1.5	4.2 ± 1.2	2.0 ± 0.5	2.3 ± 0.9	4.5 ± 0.3	274 ± 60
2 (n = 78)	7.2 ± 1.5	4.6 ± 1.3	2.0 ± 0.5	2.6 ± 1.0	4.6 ± 0.3	287 ± 64
≥3 (n = 16)	7.5 ± 1.6	5.0 ± 1.3	1.9 ± 0.4	2.7 ± 0.8	4.8 ± 0.3	288 ± 74
F	10.3	8.8	5.0	2.5	7.1	1.9
*p*	**<0.001**	**<0.001**	**0.002**	0.061	**<0.001**	0.120
MetS comp. No.	WHtR	SBP, mm Hg	FPG, mmol/L	FPI, µIU/mL	QUICKI	HDL-C, mmol/L
0 (n = 801)	0.42 ± 0.03	106 ± 8	4.6 ± 0.32	9.1 (6.0; 14.0)	0.348 ± 0.024	1.63 ± 0.25
1 (n = 336)	0.44 ± 0.05	108 ± 10	4.7 ± 0.4	10.7 (6.6; 17.4)	0.339 ± 0.026	1.33 ± 0.27
2 (n = 78)	0.49 ± 0.07	113 ± 10	4.8 ± 0.5	13.3 (7.9; 22.4)	0.329 ± 0.027	1.29 ± 0.35
≥3 (n = 16)	0.53 ± 0.08	121 ± 8	5.0 ± 0.6	21.2 (12.5; 36.1)	0.307 ± 0.023	1.24 ± 0.23
F	114	26.2	11.3	38.8	34.2	135
*p*	**<0.001**	**<0.001**	**<0.001**	**<0.001**	**<0.001**	**<0.001**
MetS comp. No.	TAG, mmol/L	AIP	UA, mmol/L	cMSS	CRP, mg/L	
0 (n = 801)	0.81 ± 0.29	−0.33 ± 0.17	254 ± 50	1.70 ± 0.29	0.5 (0.1; 1.5)	
1 (n = 336)	0.97 ± 0.49	−0.17 ± 0.19	260 ± 50	2.10 ± 0.29	0.6 (0.2; 1.9)	
2 (n = 78)	1.15 ± 0.68	−0.10 ± 0.24	272 ± 55	2.38 ± 0.42	1.2 (0.3; 3.8)	
≥3 (n = 16)	1.46 ± 0.54	0.05 ± 0.19	302 ± 47	2.77 ± 0.39	1.0 (0.3; 3.2)	
F	39.3	100	7.7	267	14.4	
*p*	**<0.001**	**<0.001**	**<0.001**	**<0.001**	**<0.001**	

MetS, metabolic syndrome; No., number; Leu, leukocytes; Neu, neutrophils; Ly, lymphocytes; NLR, neutrophils-to-lymphocytes ratio; Ery, erythrocytes, Plt, platelets; WHtR, waist-to-height ratio; SBP, systolic blood pressure; FPG, fasting plasma glucose; FPI, fasting plasma insulin; QUICKI, quantitative insulin sensitivity check index; HDL-C, high-density lipoprotein cholesterol; TAG, triacylglycerols; AIP, atherogenic index of plasma; UA, uric acid; cMSS, continuous metabolic syndrome score; CRP, C-reactive protein; F, the ratio of the between-group and within-group variation (if the null hypothesis is true, you expect F value close to 1). Data were compared using the analysis of variance. Normally distributed data are given as mean ± standard deviation (SD). Data not fitting to normal distribution were log-transformed and are presented as back-transformed geometric mean (−1 SD, +1 SD). *p* ≤ 0.05 was considered statistically significant and is shown in bold.

**Table 6 children-11-00066-t006:** Decision-tree visualizing patterns of how the presence or absence of independent categorical variables impacts leukocyte, erythrocyte, and platelet counts in males.

Node	Mean ± SD	n (%)	Parent Node	Primary Independent Variable			
				Variable	F	P_B_	Split Values
	Leukocytes						
0	6.3 ± 1.3	1188					
1	6.2 ± 1.3	1115 (93.9)	0	CRP class	390	<0.001	CRP: 0–3 mg/L
2	6.9 ± 1.4	73 (6.1)	0		390	<0.001	CRP > 3 mg/L
3	6.2 ± 1.3	980 (82.5)	1	WHtR class	310	<0.001	WHtR < 0.5
4	6.7 ± 1.1	135 (11.4)	1		310	<0.001	WHtR ≥ 0.5
5	6.8 ± 1.3	30 (2.5)	3	TAG class	133	<0.001	TAG > 1.7 mmol/L
6	6.2 ± 1.2	950 (80.0)	3		133	<0.001	TAG ≤ 1.7 mmol/L
7	6.6 ± 1.1	112 (9.4)	4	FPI class	52	<0.001	FPI < 20 µIU/mL
8	7.0 ± 1.2	23 (1.9)	4		52	<0.001	FPI ≥ 20 µIU/mL
	Erythrocytes						
0	5.15 ± 0.30	1188					
1	5.13 ± 0.30	1035 (87.1)	0	WHtR class	378	<0.001	WHtR < 0.5
2	5.25 ± 0.30	153 (12.9)	0		378	<0.001	WHtR ≥ 0.5
3	5.27 ± 0.33	31 (2.6)	1	TAG class	126	<0.001	TAG ≥ 1.7 mmol/L
4	5.13 ± 0.30	1004 (84.5)	1		126	<0.001	TAG < 1.7 mmol/L
5	5.24 ± 0.30	122 (10.3)	2	FPI class	33	<0.001	FPI < 20 µIU/mL
6	5.32 ± 0.32	31 (2.6)	2		33	<0.001	FPI ≥ 20 µIU/mL
7	5.16 ± 0.31	230 (19.4)	4	SBP class	65	<0.001	SBP ≥ 130 mm Hg
8	5.12 ± 0.29	774 (65.2)	4		65	<0.001	SBP < 130 mm Hg
9	5.45 ± 0.35	4 (0.3)	5	FPG class	45	<0.001	FPG ≥ 5.6 mmol/L
10	5.23 ± 0.29	118 (9.9)	5		45	<0.001	FPG < 5.6 mmol/L
	Platelets						
0	246 ± 48	1188					
1	244 ± 47	1035 (87.1)	0	WHtR class	378	<0.001	WHtR < 0.5
2	264 ± 47	153 (12.9)	0		378	<0.001	WHtR ≥ 0.5
3	243 ± 47	980 (82.5)	1	CRP class	126	<0.001	CRP: 0–3 mg/L
4	255 ± 47	55 (4.6)	1		126	<0.001	CRP > 3 mg/L
5	258 ± 46	107 (9.0)	2	UA class	33	<0.001	UA < 420 µmol/L
6	276 ± 49	46 (3.9)	2		33	<0.001	UA ≥ 420 µmol/L
7	236 ± 46	109 (9.2)	3	HDL-C class	65	<0.001	HDL-C < 1.03 mmol/L
8	244 ± 47	871 (73.3)	3		65	<0.001	HDL-C ≤ 1.03 mmol/L
9	251 ± 39	9 (0.8)	5	FPG class	45	0.007	FPG ≥ 5.6 mmol/L
10	259 ± 45	98 (8.2)	5		45	0.007	FPG < 5.6 mmol/L

F, ratio of the between-group and within-group variation; P_B_, Bonferroni adjusted *p*; CRP, C-reactive protein; WHtR, waist-to-height ratio; TAG, triacylglycerols; FPI, fasting plasma insulin; SBP, systolic blood pressure; FPG, fasting plasma glucose; UA, uric acid; HDL-C, high-density lipoprotein cholesterol; decision-tree algorithm: chi-squared automatic interaction detection (CHAID). An aid to reading the table, e.g., leukocyte counts: node 0 represents the whole cohort, nodes 1 and 2 represent splits of the cohort according to dichotomic variable characterized by split values. Nodes 1 and 2 thus represent splits according to the CRP cut-offs, yielding node 1 with CRP up to 3 mg/L and node 2 including males with CRP > 3 mg/L. Nodes 3 and 4 represent splits of parent node 1 (e.g., males with CRP levels up to 3 mg/L) according to the presence (node 4) or absence (node 3) of central obesity. Nodes 5 and 6 stem from node 3 (centrally lean males) which is split according to TAG levels below/above the cut-off point, and nodes 7 and 8 represent insulin-sensitive and insulin-resistant, respectively, centrally obese males displaying CRP levels below 3 mg/L.

**Table 7 children-11-00066-t007:** Decision-tree visualizing patterns of how the presence or absence of independent categorical variables impacts leukocyte, erythrocyte, and platelet counts in females.

Node	Mean ± SD	n (%)	Parent Node	Primary Independent Variable			
				Variable	F	P_B_	Split Values
	Leukocytes						
0	6.6 ± 1.5	1231					
1	6.6 ± 1.5	1106 (89.8)	0	CRP class	747	<0.001	CRP: 0–3 mg/L
2	7.5 ± 1.5	125 (10.2)	0		747	<0.001	CRP > 3 mg/L
3	7.1 ± 1.4	89 (7.2)	1	WHtR class	236	<0.001	WHtR ≥ 0.5
4	6.5 ± 1.4	1017 (82.6)	1		236	<0.001	WHtR < 0.5
5	7.9 ± 1.5	33 (2.7)	2	HDL-C class	60	<0.001	HDL-C < 1.29 mmol/L
6	7.3 ± 1.5	92 (7.5)	2		60	<0.001	HDL-C ≤ 1.29 mmol/L
7	7.0 ± 1.4	38 (3.1)	4	TAG class	97	<0.001	TAG ≥ 1.7 mmol/L
8	6.5 ± 1.4	979 (79.5)	4		97	<0.001	TAG < 1.7 mmol/L
	Erythrocytes						
0	4.55 ± 0.28	1231					
1	4.62 ± 0.32	111 (9.0)	0	WHtR class	378	<0.001	WHtR ≥ 0.5
2	4.55 ± 0.28	1120 (91.0)	0		378	<0.001	WHtR < 0.5
3	4.73 ± 0.31	16 (1.3)	1	UA class	126	<0.001	UA ≥ 340 µmol/L
4	4.60 ± 0.32	95 (7.7)	1		126	<0.001	UA < 340 µmol/L
5	4.66 ± 0.29	21 (1.7)	2	FPG class	33	<0.001	FPG ≥ 5.6 mmol/L
6	4.54 ± 0.28	1099 (89.3)	2		33	<0.001	FPG < 5.6 mmol/L
7	4.52 ± 0.26	223 (18.1)	6	HDL-C class	65	<0.001	HDL-C < 1.29 mmol/L
8	4.55 ± 0.28	876 (71.2)	6		65	<0.001	HDL-C ≤ 1.29 mmol/L
	Platelets						
0	274 ± 56	1231					
1	304 ± 63	62 (5701)	0	TAG class	363	<0.001	TAG ≥ 1.7 mmol/L
2	272 ± 55	1169 (95.0)	0		363	<0.001	TAG < 1.7 mmol/L
3	272 ± 55	1154 (93.7)	2	SBP class	151	<0.001	SBP < 130 mm Hg
4	314 ± 86	15 (1.2)	2		151	<0.001	SBP ≥ 130 mm Hg
5	271 ± 54	1050 (85.3)	3	CRP class	103	<0.001	CRP: 0–3 mg/L
6	284 ± 59	104 (8.4)	3		103	<0.001	CRP > 3 mg/L

F, ratio of the between-group and within-group variation; P_B_, Bonferroni adjusted *p*; CRP, C-reactive protein; WHtR, waist-to-height ratio; HDL-C, high-density lipoprotein cholesterol; TAG, triacylglycerols; UA, uric acid; FPG, fasting plasma glucose; SBP, systolic blood pressure; decision-tree algorithm: chi-squared automatic interaction detection (CHAID).

## Data Availability

The data presented in this study are available on request from the corresponding author. The data are not publicly available due to due to privacy concerns.

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
