# Peer review of "Association of Leukocyte, Erythrocyte, and Platelet Counts with Metabolic Syndrome and Its Components in Young Individuals without Overt Signs of Inflammation: A Cross-Sectional Study"

_children, 2024, doi:10.3390/children11010066_

Round 1

Reviewer 1 Report

Comments and Suggestions for Authors

In the present manuscript, Šebeková et al. assessed whether metabolic syndrome affected blood cell counts in participants aged 16-20 years. The study involved 1188 male and 1231 female participants, without any evident signs of inflammation, from the Bratislava Region, in Slovakia. Among other observations, metabolic syndrome increased the leukocyte, erythrocyte, and platelet counts.

Despite representing only one relatively small geographical location, this study is robust, due to the large number of participants. It contributes to a better understanding of the impact of metabolic syndrome on young patients. Considering the increasing incidence of metabolic syndrome in children, adolescents, and young adults, this work is timely and could be clinically relevant.

Specific remarks:

1. In the abstract, line 19, please include the full name of CRP.

2. Throughout the manuscript, please use only one acronym for metabolic syndrome (either MS or MetS).

3. In Table 2, the participant numbers (males n=1203; females n=1291) are not in agreement with those indicated in the rest of the manuscript. Please verify.

4. Please consider adding an acknowledgments section, thanking the many volunteer participants for their contribution to this study.

5. Concerning the ethics committee approval of the study, please include the approval code, if available.

6. In line 303, please check if "Observing" should be corrected to "Observed".

7. In line 18, please correct "correlated significantly" to "significantly correlated". In line 64, please correct "addressing simultaneously" to "simultaneously addressing". In line 71, please correct "evaluated separately" to "separately evaluated". In line 377, please correct "As shown formerly" to "As formerly shown".

Comments on the Quality of English Language

Please check remarks #6 and #7.

Author Response

In the present manuscript, Šebeková et al. assessed whether metabolic syndrome affected blood cell counts in participants aged 16-20 years. The study involved 1188 male and 1231 female participants, without any evident signs of inflammation, from the Bratislava Region, in Slovakia. Among other observations, metabolic syndrome increased the leukocyte, erythrocyte, and platelet counts.

Despite representing only one relatively small geographical location, this study is robust, due to the large number of participants. It contributes to a better understanding of the impact of metabolic syndrome on young patients. Considering the increasing incidence of metabolic syndrome in children, adolescents, and young adults, this work is timely and could be clinically relevant.

We are thankful to the reviewer for spending her/his valuable time to help us improve our paper and for constructive suggestions, which were, to our best, implemented in the revised paper highlighted in red).

Specific remarks:

  1. In the abstract, line 19, please include the full name of

Thanks for the pointing out, we replaced the abbreviation with the full name.

  1. Throughout the manuscript, please use only one acronym for metabolic syndrome (either MS or MetS).

We apologize for the inconsistent abbreviations; we corrected them in the revised paper. We use MetS throughout the manuscript.

  1. In Table 2, the participant numbers (males n=1203; females n=1291) are not in agreement with those indicated in the rest of the Please verify.

Many thanks for pointing out this discrepancy; we corrected the number of participants in the revised paper.

  1. Please consider adding an acknowledgments section, thanking the many volunteer participants for contributing to this study.

We omitted the acknowledgment since ICMJE states, " editors are advised to require that the corresponding author obtain written permission to be acknowledged from all acknowledged individuals." However, we are aware of the role, engagement, and cooperation of parents and their children, school directors and teachers, staff members engaged in measurements and recordings; and the realization team of the Respect for Heath Study. Thus, as suggested, we included the acknowledgment, albeit obtaining permission from those eligible to be acknowledged is illusory.

  1. Concerning the ethics committee approval of the study, please include the approval code, if

The approval was not coded; it is identifiable only by the name of the study and the date of its approval. In Slovakia, it is not the duty of the Ethics Boards to code the approvals – we often have problems explaining this practice.

  1. In line 303, please check if "Observing" should be corrected to "Observed".

As suggested, we corrected the phrasing.

  1. In line 18, please correct "correlated significantly" to "significantly correlated". In line 64, please correct "addressing simultaneously" to "simultaneously addressing". In line 71, please correct "evaluated separately" to "separately evaluated". In line 377, please correct "As shown formerly" to "As formerly shown".

Many thanks for pointing out our incorrectly used word ordering; as suggested, we corrected it in the revised manuscript.

Reviewer 2 Report

Comments and Suggestions for Authors

The subject is widely recognized and extensively researched. The findings presented, however, do not introduce any novel contributions to the field. I suggest incorporating a new parameter, revising the article, and then resubmitting it for consideration.

Author Response

The subject is widely recognized and extensively researched. The findings presented, however, do not introduce any novel contributions to the field. I suggest incorporating a new parameter, revising the article, and then resubmitting it for consideration. We thank the reviewer for her/his constructive suggestions to improve our paper. However, we disagree with the stated reservations.
1/ The association of the MetS and its components with blood counts in apparently healthy juveniles can hardly be considered extensively researched. The reports in juveniles focus on a single blood count component (except for the study of Mansourian et al., 2014, who concurrently present data on leukocytes and erythrocyte counts). Analyses focused on a single component do not allow for the conclusion that all three major blood picture components a/ differ between MetS-positive and MetS-free subjects and that b/ they simultaneously display similar trends across the rising number of MetS components. We are unaware of any study concurrently analyzing leukocyte, erythrocyte, and platelet counts association with MetS and its components.
According to the census in 2011, the Slovak Republic had slightly below 5 400 000 inhabitants. We suppose that the relationship between the number of citizens and the number of students included in our study is acceptable. Moreover, albeit only students attending the state secondary schools in Bratislava Region were examined, Bratislava is the capital with several highly specialized secondary schools. We have no data on the proportion of second/third generation participants born in the region, students staying during the school year with their relatives, in boarding schools, or daily traveling to Bratislava from other regions of Slovakia, or those living abroad, e.g., dialy crossing from Hungary or Austria, as “the boarders” (we are ib EU) are situated directly within Bratislava. 2/"The findings presented, however, do not introduce any novel contributions to the field". Except for the abovementioned facts, we aimed to characterize phenotypes associated with the highest leukocyte, erythrocyte, and platelet counts. We are not aware of any study attempting to analyze similar characteristics. 3/ We are thankful to the reviewer for giving us a free hand to decide which new parameter should be incorporated. - Hematologic variables: The Children Journal is not a focused hematologic paper. We supposed that extensive reporting on differential blood count variables is beyond the journal's scope. However, the reviewer perhaps considered differential white blood cell counts of clinical impact; thus, we included the neutrophils-to-lymphocytes ratio in the revised paper. We selected NLR, as it has been proposed as a marker of low-grade chronic inflammation in cases when leukocyte counts are within the reference range. - Other metabolic variables, e.g., those not included in the MetS components: we carefully selected two variables with potential pathophysiological links with leukocyte or erythrocyte counts (CRP, uric acid). Comparison of their trends to those of particular MetS components and blood counts across the rising number of MetS components provides a clear picture of the degree of associations. We suppose that quoting several other biochemical variables with known positive or negative relationships with the increasing number of MetS components or the severity of the MetS would not add any essential additional information.
- Other new parameters: the impact of genetics and lifestyle factors on both blood counts and MetS components, or MetS severity, is widely recognized and extensively studied. Unfortunately, we have no data on our students' genetics, dietary regimen, behavioral patterns, or physical activity. We suppose that only incorporating this information could lead to the required substantial article revision before its resubmission. We comment on these weaknesses of our contribution in the revised manuscript, hoping that other groups that have information missing from us could fill the informational gap.

Reviewer 3 Report

Comments and Suggestions for Authors

Review of the article "Association of leukocyte, erythrocyte, and platelet counts with metabolic syndrome and its components: A cross-sectional study in young individuals without overt signs of inflammation" by Katarína Šebeková, Radana Gurecká, and Ľudmily Podracká:

Strengths:

The paper uses a large dataset of 1188 men and 1231 women aged 16-20 years, making the results more representative of the population of young individuals. The authors performed a comprehensive analysis, including leukocytes, erythrocytes and platelets, and assessed the association of these parameters with metabolic syndrome (MetS) and its components. The results indicate that the number of blood cells increases with the number of MetS components manifested, which may be important for understanding the relationship between metabolic status and hematological parameters. The paper attempts to characterize the phenotypes associated with high blood cell counts, which may be relevant for further research on the relationship between hematological changes and cardiovascular disease risk. The authors suggest that an increase in normal blood cell counts may have prognostic significance, which opens the field for further research.

Limitations:

The main limitation is the cross-sectional nature of the study, making it impossible to determine the causal direction between metabolic syndrome and blood cell counts. Only inferences about associations are possible. The results of the study are limited to young participants, which may introduce some limitations in generalizing the results to the general population. The study does not take into account possible comorbidities or factors affecting the results, which may introduce some inaccuracy. The data are based on single measurements and blood samples from each participant, which may affect the precision of the results. The study focuses only on students attending high schools in the Bratislava region, which may limit the overall representativeness of the results.

Recommendations to the authors:

The authors could consider taking into account other factors, such as lifestyle or genetics, that may influence the association between hematological parameters and metabolic syndrome.

Longitudinal studies are recommended to more precisely understand the relationship between blood cell count and cardiovascular disease risk.It could be valuable to extend the statistical analysis to more specifically identify subgroups susceptible to MetS-related hematological changes.The study could be enriched by analyzing the influence of environmental factors, such as diet and physical activity.It is recommended to rewrite the conclusions, which should not include references to the literature. It is noteworthy that despite its limitations, this work makes an important contribution to understanding the relationship between hematological parameters and metabolic syndrome in young people.

Author Response

Review of the article "Association of leukocyte, erythrocyte, and platelet counts with metabolic syndrome and its components: A cross-sectional study in young individuals
without overt signs of inflammation" by Katarína Šebeková, Radana Gurecká, and Ľudmily Podracká:
Strengths:
The paper uses a large dataset of 1188 men and 1231 women aged 16-20 years, making the results more representative of the population of young individuals. The authors performed a comprehensive analysis, including leukocytes, erythrocytes and platelets, and assessed the association of these parameters with metabolic syndrome (MetS) and its components. The results indicate that the number of blood cells increases with the number of MetS components manifested, which may be important for understanding the relationship between metabolic status and hematological parameters. The paper attempts to characterize the phenotypes associated with high blood cell counts, which may be relevant for further research on the relationship between hematological changes and cardiovascular disease risk. The authors suggest that an increase in normal blood cell counts may have prognostic significance, which opens the field for further research.
We are thankful to the reviewer for spending her/his valuable time to help us improve our paper and for constructive suggestions, which were, to our best, implemented in the revised paper (highlighted in blue).
Limitations:
The main limitation is the cross-sectional nature of the study, making it impossible to determine the causal direction between metabolic syndrome and blood cell counts. Only inferences about associations are possible. The results of the study are limited to young participants, which may introduce some limitations in generalizing the results to the general population. The study does not take into account possible comorbidities or factors affecting the results, which may introduce some inaccuracy. The data are based on single measurements and blood samples from each participant, which may affect the precision of the results. The study focuses only on students attending high schools in the Bratislava region, which may limit the overall representativeness of the results.
We are aware of the mentioned limitations. Except for those mentioned in the manuscript, we added that we have no data on the students' genetics, family anamnesis, and lifestyle factors, which could affect both the manifestation of the MetS components and blood counts.
We are also aware of the fact that we do not present a representative study on 16-to-20-year-old Slovaks. However, Bratislava is the capital with several unique, specialized secondary schools. We have no information on whether students attending the secondary schools in the Bratislava Region were permanent residents or whether they were staying with relatives or at boarding schools during the school year.
Recommendations to the authors:
The authors could consider taking into account other factors, such as lifestyle or genetics, that may influence the association between hematological parameters and metabolic syndrome.
As suggested, we included the mentioned limitation in the revised manuscript.
Longitudinal studies are recommended to more precisely understand the relationship between blood cell count and cardiovascular disease risk.It could be valuable to extend the statistical analysis to more specifically identify subgroups susceptible to MetS-related hematological changes.The study could be enriched by analyzing the influence of environmental factors, such as diet and physical activity.It is recommended to rewrite the conclusions, which should not include references to the literature. It is noteworthy that despite its limitations, this work makes an important contribution to understanding the relationship between hematological parameters and metabolic syndrome in young people.
Many thanks for the suggestions. Unfortunately, we have no data on additional factors that could help to identify susceptible subgroups. This fact is mentioned among the limitations in the revised manuscript.
As recommended, we modified the conclusions. We also transferred the sentences with citations, to the discussion.

Round 2

Reviewer 2 Report

Comments and Suggestions for Authors

Accept in present form.